# COVID-19 in Italy: Is the Mortality Analysis a Way to Estimate How the Epidemic Lasts?

**DOI:** 10.3390/biology12040584

**Published:** 2023-04-11

**Authors:** Pietro M. Boselli, Jose M. Soriano

**Affiliations:** 1Group of Nutritional Modelling Biology, Departament de Biosciencies, University of Milan, 20122 Milan, Italy; 2Food & Health Lab, Institute of Materials Science, University of Valencia, 46980 Paterna, Spain; 3Joint Research Unit on Endocrinology, Nutrition and Clinical Dietetics, Health Research Institute La Fe-University of Valencia, 46026 Valencia, Spain

**Keywords:** coronavirus data analysis, COVID-19, duration of the epidemic, Italy, mortality curve

## Abstract

**Simple Summary:**

Every epidemic generates a series of problems of a health, economic, social, and environmental nature at a local and/or planetary level. It is essential to identify them, understand their dynamic evolution, and then find a solution as soon as possible. In this study, a simple mathematical model was used to interpret the temporal trends of the positive-alive and the dead. The purpose of this study is to find a way to predict, if possible, the duration of the epidemic and its phases. Obviously, the epidemic will last until the number of positive-alive collapses towards zero, and that of the accumulated dead will stabilize at the maximum value. The analysis was conducted in Italy in the period between January 2020 and December 2022. The results obtained show that both the analyzes of the positive-alive and dead curves provide reliable predictions that are consistent with each other. However, the analysis of cumulative deaths leads to more precise forecasts of the duration of both the phases and the entire development of the epidemic.

**Abstract:**

When an epidemic breaks out, many health, economic, social, and political problems arise that require a prompt and effective solution. It would be useful to obtain all information about the virus, including epidemiological ones, as soon as possible. In a previous study of our group, the analysis of the positive-alive was proposed to estimate the epidemic duration. It was stated that every epidemic ends when the number of positive-alive (=infected-healed-dead) glides toward zero. In fact, if with the contagion everyone can enter the epidemic phenomenon, only by healing or dying can they get out of it. In this work, a different biomathematical model is proposed. A necessary condition for the epidemic to be resolved is that the mortality reaches the asymptotic value, from there, remains stable. At that time, the number of positive-alive must also be close to zero. This model seems to allow us to interpret the entire development of the epidemic and highlight its phases. It is also more appropriate than the previous one, especially when the spread of the infection is so rapid that the increase in live positives is staggering.

## 1. Introduction

On 6:06 a.m. CET, 21 March 2023, the World Health Organization (WHO) have been 761,071,826 confirmed coronavirus disease 2019 (COVID-19), including 6,879,677 deaths. Furthermore, as of 21 March 2023, a total of 13,260,401,200 vaccine doses have been administered [1]. Nowadays, several researchers are searching for antique and new tools in data analysis to modeling and forecasting studies applied in this pandemic. The epidemic analysis for forecasting COVID-19 using, among others, the susceptible, infected, and recovered (SIR) [2], susceptible, infected, recovered, and deceased (SIRD) [3], susceptible, exposed, infected, and recovered (SEIR) [4], susceptible, exposed, infected, recovered and dead (SEIRD) [5], SEIRD model with the compartment of vaccinated people (SEIRDV) [6], and Moving Average [7] models, or even hybrid dynamic model as is SEIRD with Automatic Regressive Integrated Moving Average (ARIMA) corrections [8] or data-driven hybrid technique by integrating an ensemble empirical mode decomposition (EEMD), an autoregressive integrated moving average (ARIMA), with a nonlinear autoregressive artificial neural network (NARANN), called the EEMD-ARIMA-NARANN model [9]. Predictive models of mortality have also been studied, highlighting the study of Friedman et al. [10], which observed that seven COVID-19 models covered more than five countries, suggesting that effects of seasonality or continued slow [11,12,13,14,15,16,17] declines in mortality could be responsible for converging in their predictions for the June–August 2020 period. Lately, a particular turning point has become apparent when it comes to studies focusing on the end of COVID-19 using mathematical models. In fact, The Independent Panel for Pandemic Preparedness and Response [18] has urged WHO to develop a road map to guide efforts towards ending the COVID-19 pandemic within countries and globally. Some countries [19,20] are focusing on the development of the model to know the end of this pandemic. Heywood and Macintyre [21] indicated that the elimination of COVID-19 or any infectious disease requires evidence of the maintenance of a basic reproduction number (R 0) below 1 in a health system with the capacity to detect a case of infection if it does occur. When a disease occurs, especially if it is infectious, one wonders how long it will last. It is important to give an answer to know how much time we have available to organize preventive measures and effective treatments. In our previous publication, Boselli et al. [22] proposed the mathematical analysis of the positive-alive trend as a method to try to predict its duration. At the beginning of an epidemic, the number of live positives increases until it reaches a maximum value and then glides towards a negligible value, for which the epidemic phenomenon can be considered concluded. However, the analysis of positive-alive has become unsuitable to describe the epidemic over time because it developed results in successive phases. The aim of this work is to estimate how long the entire epidemic process could last in Italy.

## 2. Materials and Methods

### 2.1. Data

The cumulative data on the number of infected, healed, and dead in the period from 1 January 2020 to 31 December 2022 had been collected on a daily basis (from official sources, e.g., Istituto Superiore di Sanità, control and surveillance) [23]. Then, the data were reorganized in a new table that reports both the percentage of positives-alive and that of the dead; both referred to the Italian population (Table 1) and occurred in a set time interval [24]. It should be remembered that the number of living positives is calculated as the difference between the number of infected and the sum of the recovered and the deaths. The percentage of positive-alive is given by the ratio between the number of positive-alive and the number corresponding to one-hundredth of the population. Furthermore, the percentage of deaths, obtained from the ratio between the number of deaths and one-hundredth of the population in the same time interval, is equal to the value of Mortality [25].

### 2.2. Model

It is assumed that the end of the epidemic will occur when the Mortality index reaches a numerical value very close (for example, ≥95%) to the asymptotic one and will keep it stable over time. The choice to analyze the trend of mortality to try to predict the duration of the epidemic was made on the basis of some evidence:The trend of mortality (observed data) develops in phases;The trend of positive-alive (observed data) shows some peaks from mid-2021;Since the overall outcome of an epidemic depends on the number of deaths, mortality analysis could lead to predicting its duration.

Italian population in 2021: 59,258,000.

Therefore, it is reasonable to draw this simple Figure 1:

The model is described by the following system of differential Equations:$$\left\{\begin{array}{l} dQ/dt=-k_{1}Q\\ dPA/dt=-k_{2}PA+k_{1}Q\\ dM/dt=+k_{2}PA\\ dH/dt=+k_{3}PA \end{array}\right.$$

The formal solution of the system consists of the following integral Equations:Q(t) = Q_0_ * [EXP(−k_1_*t)]
PA(t) = [k_1_Q_0/_(k_1_ − K)] * [EXP(−K*t) − EXP(−k_1_*t)]
M(t) = [k_2_Q_0_/K] + [k_2_Q_0_/(k_1_ − K)] * [EXP(−k_1_*t)] − [k_1_k_2_Q_0_/K(k_1_ − K)] * [EXP(−K*t)]
H(t) = [k_3_Q_0_/K] + [k_3_Q_0_/(k_1_ − K)] * [EXP(−k_1_*t)] − [k_1_k_3_Q_0_/K(k_1_ − K)] * [EXP(−K*t)]

In which:

t is the number of months that have passed since the beginning of the epidemic

Q_0_ = 100% is the entire population at the beginning

k_1_ is the speed of entry into PA

K = (k_2_ + k_3_) is the exit speed from PA

k_2_ is the entry speed into M

k_3_ is the entry speed into H.

## 3. Results

The observed % values of positive-live and deaths are both graphed up to the end of 2022 (Figure 1).

The values k_1_, k_2_, K, and the asymptote k_2_/K were calculated from the Mortality equation (see 2.2 Model—The formal solution) by best fitting the data using the Ordinary Least Squares method (OLS) (Table 2).

The best-fitting curve of the observed mortality is reported (Figure 2). To make the mortality curve more understandable, the full scale of the y-ordinate has been changed.

The incremental values of Mortality in the individual phases of the epidemic are reported (Table 3).

Three hypotheses of the duration of the epidemic, corresponding to the achievement of 95, 98, and 99% of asymptotic mortality, are shown in Table 4.

## 4. Discussion

The mathematical model, designed according to the logic of the epidemic phenomenon, was a simple tool that made it possible to achieve the purpose of the work. Of course, there are many variables and numerous contributing factors determining the trend of mortality. However, they are phenomenologically understood in the temporal development of the compartments. The trend of mortality, being terminal, summarizes them. The addition of compartments would not lead to further information due to that it would increase the constraints, making the system more rigid and difficult to solve it.

Three stages of development are reflected in Figure 1 and, more clearly, in Figure 2. The first phase, which ended in September 2020, was characterized by 16,400 deaths/month up to June and about 650 deaths/month in the following three months. This decrease was due, above all, to the lockdown carried out between March and June. At the beginning of the second phase, November 2020, an almost equal increase in Mortality was achieved (16,950 deaths/month). It stabilized throughout the summer, with about 530 deaths per month. The fairly stable mortality between June and November was most likely attributable to the outcome of the mass vaccination campaign. It is interesting to note that the first two phases have a very similar pattern, although the first develops for five months and the second for nine. Finally, at present, the third phase is underway, which began in December 2021, concurrently with the Omicron variant. It seems to constitute the final phase of the logistic curve of Mortality in the whole epidemic. It shows a slow and gradual increase, slower than in the previous two phases. In January 2022, the number of deaths was 9600; in June, 355; in December, 2370. It must be noted that, at the beginning of each phase, the period of increase in mortality perfectly overlaps with the trend of positive-alive. It seems to coincide and overlap with the appearance of new variants. The moments of stability at the end of each phase seem to signal an appropriate and effective system of prevention and treatment. Our model allows you to quickly assess the historical robustness of our predictions, which is a key factor suggested by Friedman et al. [10] for COVID-19 decision-making. The value of Mortality is more certain and less prone to errors. There may be delays in reporting the deaths that have occurred, but the number is certain. Secondly, the population does not suddenly change significantly from one year to the next. The cumulative number of deaths can only increase or remain at the maximum value reached if the measures taken to prevent contagion and treat the disease are fully effective. However, this can never happen at the beginning of the phenomenon, especially if it is a new virus of which neither the characteristics nor the methods of treatment are known. This is why the Mortality Index at the beginning makes us understand how deadly the new virus is or not. We can ask ourselves: what are the preventive and curative measures used for? Certainly not to bring the dead back to life, but to bring to life those who are not yet dead. That is, to reduce the rate of growth in the number of deaths. Among other things, the variation in the slope (speed) of the mortality curve shows us how effective the prevention and treatment system are or not. As can be seen in Figure 2, the integral mortality equation predicted, as early as January 2022, the cumulative number of deaths that actually occurred with an average error of 0.15%. The review of Friedman et al. [10] demonstrated that three models, including Delphi [11], LANL [12], and UCLA-ML [13] from October underestimated mortality, obtaining median percent errors of −9.2%, −9.1 and −10.2% at 6 weeks, respectively, but the remaining models [14,15,16,17] were relatively unbiased.

In order to estimate the duration of an epidemic, both methods of analysis seem valid because they are based on obvious postulates: the epidemic ends when the positive-alive is reduced to zero and when the Mortality reaches the maximum value and keeps it stable. There remains the opportunity to choose one method, the other, or both depending on the conditions determined by the virus, its variants, diffusibility, contagiousness, the measures to prevent its contagion, and the effectiveness of the specific therapies adopted. How is it possible to calculate the duration of an epidemic using a logistic function, which tends to the asymptote in a time tending to infinity? Then, it is necessary to choose a conventional value of Mortality sufficiently close to the asymptotic value. Among the three options, the duration of the epidemic (Table 4) could be chosen as 57 months, corresponding to the achievement of 95% of the asymptote of Mortality. Moreover, at that time, the number of daily deaths would be similar to the number of deaths from non-COVID-19 pneumonia.

## 5. Conclusions

In conclusion, it seems there are no limits to the application of mathematical models in every sector of knowledge. In particular, they are a fundamental tool for understanding all biological phenomena. The analysis of the trends Mortality curve, as well as providing various information about the characteristics of the virus, makes it possible to formulate a hypothesis of the duration of the whole phenomenon. To date, what better method could estimate the duration of a phenomenon that ends when time tends to infinity? It is also much more useful than the one already carried out on the positive-alive curve in the previous two phases. The downside is that you need to have more data available. Instead, the analysis of live positives is advantageous for shorter time intervals, for low diffusion and contagiousness values. Given the interest aroused by being able to predict the duration of an epidemic–pandemic, the applicability of the proposed model should also be studied in other environmental and territorial contexts of other regions, using national, continental, and global data.

## Data Availability

Official source from Istituto Superiore di Sanità, control and surveillance which is available in https://www.salute.gov.it/portale/nuovocoronavirus/dettaglioContenutiNuovoCoronavirus.jsp?area=nuovoCoronavirus&id=5351&lingua=italiano&menu=vuoto (accessed on 25 March 2023).

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
