# Peer review of "COVID-19 in Italy: Is the Mortality Analysis a Way to Estimate How the Epidemic Lasts?"

_biology, 2023, doi:10.3390/biology12040584_

Round 1

Reviewer 1 Report

In this work, author proposed an ordinary differential equation that account for infected, positive, mortality and healing populations in the model. Authors tried to use this model to estimate how COVID-19 epidemic can last using Italy as a case study. Authors conclude that the model allows to interpret the entire development of the epidemic and highlight phases.

In my opinion, this work is novel and interesting but need improvement in many parts.

I was wondering why authors did not include in their model what account for variants of the disease and interventions like vaccination which influences the spread of the disease and mortality due to the disease. I do not think we can appropraitely account for when the epidemic will last if we don't take these factors into consideration.

Figure 2 is not clear. Authors should represent it. Readers will find it hard to interpret this result.

There are repetitions in line 101-108

COVID-19 should be used throughout the manuscript instead of covid19, Covid etc. for uniformity sake.

Author Response

Reviewer’s comment: In this work, author proposed an ordinary differential equation that account for infected, positive, mortality and healing populations in the model. Authors tried to use this model to estimate how COVID-19 epidemic can last using Italy as a case study. Authors conclude that the model allows to interpret the entire development of the epidemic and highlight phases. In my opinion, this work is novel and interesting but need improvement in many parts.

Author’s comment: Thank you for your comment. Now, we have think that it is improved with reviewers’ comments.

Reviewer’s comment: I was wondering why authors did not include in their model what account for variants of the disease and interventions like vaccination which influences the spread of the disease and mortality due to the disease. I do not think we can appropraitely account for when the epidemic will last if we don't take these factors into consideration.

Author’s comment: According to your comment, we have added in the discussion a paragraph to clarify it. Basically, the mathematical model, designed according to the logic of the epidemic phenomenon, was a simple tool that made it possible to achieve the purpose of the work. Of course, there are many variables and numerous are the contributing factors that determine the trend of mortality. However, they are phenomenologically understood in the temporal development of the compartments. The trend of mortality, being terminal, summarizes them. The addition of compartments would not lead to further information due to that it would increase the constraints, make the system more rigid and difficult to solve it.

Reviewer’s comment Figure 2 is not clear. Authors should represent it. Readers will find it hard to interpret this result.

Author’s comment: According to your comment, we have clarified it in the results section and discussed in the discussion section.

Reviewer’s comment: There are repetitions in line 101-108

Author’s comment: According to your comment, these lines have been deleted.

Reviewer’s comment: COVID-19 should be used throughout the manuscript instead of covid19, Covid etc. for uniformity sake.

Author’s comment: According to your comment, we have changed it, in the manuscript.

Reviewer 2 Report

The manuscript entitled "COVID-19 in Italy: Is the Mortality Analysis a Way to Estimate How the Epidemic Last?" present a novel biomathematical model that offers a comprehensive framework for interpreting the entire trajectory of an epidemic based on the positive-alive and dead curves.

I have carefully evaluated your paper and have identified a point that requires further clarification.

Specifically, it is not clear how your proposed model differs from the models discussed in the paper "Predictive performance of international COVID-19 mortality forecasting models" by Friedman et al. Therefore, I would like to request that you provide a more detailed explanation of the distinctions between your model and the models outlined in Friedman et al.'s paper.

https://doi.org/10.1038/s41467-021-22457-w

Furthermore, I recommend that you introduce some of the models discussed in Friedman et al.'s paper, as well as their respective references, in your Introduction section. This will help contextualize your proposed model within the existing literature and make your paper more informative for readers.

In addition, I would also like to emphasize the importance of discussing and comparing your findings with the models presented in Friedman et al.'s paper in the Discussion. It is essential to contextualize your proposed model within the broader literature and clearly demonstrate the unique contributions of your work in advancing the field.

Minor point: 
Line 102 - term popolazione should be population
Delete lines 105 to 108. Same text on the previous lines.

Best Regards.

Author Response

Reviewer’s comment: The manuscript entitled "COVID-19 in Italy: Is the Mortality Analysis a Way to Estimate How the Epidemic Last?" present a novel biomathematical model that offers a comprehensive framework for interpreting the entire trajectory of an epidemic based on the positive-alive and dead curves. I have carefully evaluated your paper and have identified a point that requires further clarification.

Author’s comment: Thank you for comment and new ideas.

Reviewer’s comment: Specifically, it is not clear how your proposed model differs from the models discussed in the paper "Predictive performance of international COVID-19 mortality forecasting models" by Friedman et al. Therefore, I would like to request that you provide a more detailed explanation of the distinctions between your model and the models outlined in Friedman et al.'s paper. https://doi.org/10.1038/s41467-021-22457-w

Author’s comment: According to your comment, we have added in the manuscript.

Reviewer’s comment: Furthermore, I recommend that you introduce some of the models discussed in Friedman et al.'s paper, as well as their respective references, in your Introduction section. This will help contextualize your proposed model within the existing literature and make your paper more informative for readers.

Author’s comment: According to your comment, we have added it to clarify the manuscript.

Reviewer’s comment: In addition, I would also like to emphasize the importance of discussing and comparing your findings with the models presented in Friedman et al.'s paper in the Discussion. It is essential to contextualize your proposed model within the broader literature and clearly demonstrate the unique contributions of your work in advancing the field.

Author’s comment: According to your comment, we have added this idea in introduction and discussion sections. Thank you for comment which help to clarify the manuscript.

Reviewer’s comment: Minor point:  Line 102 - term popolazione should be population

Author’s comment: According to your comment, this term has been changed.

Reviewer’s comment: Delete lines 105 to 108. Same text on the previous lines.

Author’s comment: According to your comment, these lines have been deleted.

Round 2

Reviewer 1 Report

No further comments for authors.

Reviewer 2 Report

Dear authors, thank you for submit the updated version of manuscript the manuscript and answered all the questions.